# Relationship Between Dietary Habits and Stress Responses Exerted by Different Gut Microbiota

**DOI:** 10.3390/nu17081388

**Published:** 2025-04-20

**Authors:** Kouji Satoh, Makoto Hazama, Mari Maeda-Yamamoto, Jun Nishihira

**Affiliations:** 1Department of Medical Management and Informatics, Hokkaido Information University, Ebetsu 069-0832, Japan; ks46317@do-johodai.ac.jp (K.S.); m-hazama@do-johodai.ac.jp (M.H.); 2Institute of Food Research, National Agriculture and Food Research Organization, Tsukuba 305-8642, Japan; marimy@affrc.go.jp

**Keywords:** gut microbiota, stress response, dietary habitat, enterotype

## Abstract

Background/Objectives: A number of studies have reported on the improvement in physical and psychological diseases through diet; however, the findings for these ameliorative effects have differed. Such differences may be due to the varying metabolism of the nutrient content in food among subjects. It has been reported that differences in the enterotypes of gut microbiota are associated with metabolic differences, and enterotypes vary between countries and regions. This study investigated whether differences in gut microbiota affect the relationship between dietary habits and stress responses. Methods: We administered a questionnaire to 810 subjects who participated in the “Sukoyaka Health Survey” regarding their dietary habits and stress reactions. We also performed an analysis of the gut microbiota from fecal samples. Results: The gut microbiota was grouped into four clusters based on the abundance of genus strains. The relationship between dietary habits and stress responses revealed two patterns of eating: one where more frequent intakes were associated with a lower stress response, and another with a higher stress response. We investigated the relationship between dietary habits and stress responses for each gut microbiota cluster. The results showed that the relationship between dietary habits and stress responses differed for each cluster. Conclusions: Our analysis showed that dietary habits affect stress responses, but the relationship varies depending on the gut microbiota. This finding suggests that one of the factors for the difference in the ameliorative efficacy of physical and psychological diseases through diet is the difference in the abundance ratio of the gut microbiota (enterotype).

## 1. Introduction

Numerous studies have found that gut bacteria are important drivers of health conditions and diseases such as obesity, rheumatism, colitis, and cancer. The relationship between the gut microbiota and brain function is also known to be related to stress and neurological diseases such as Alzheimer’s and Parkinsonian disorders [1,2,3]. The implications of gut microbiota for health vary according to diet, lifestyle, age, sex, and medical history [4,5,6]. Through the large-scale analysis of the gut microbiota, enterotype is classified into two or three types for strains with high abundance at the genus level [7,8]. Two major enterotypes among Japanese individuals have been reported to be Bacteroides and Prevotella [9]. Bacteroides-type diets are characterized by a high intake of lipids and proteins, while Prevotella-type diets are characterized by a high intake of dietary fiber. Differences in enterotype may thus be attributed to differences in long-term dietary habits [6]. Short-term dietary habits lead to changes in the abundance of gut microbiota but have no major effect on changes in enterotype [6]. Differences in enterotype are also reported to be involved in the metabolic functions of foods, not merely differences in gut microbiota [10,11,12].

On the other hand, stress is a cause of various diseases, especially mental disorders. Mental illness affects one in eight individuals worldwide, with many affected by anxiety disorders and depression [13]. Various mental illnesses caused by stress greatly impact social and economic activities, with economic losses exceeding healthcare costs [14]. Therefore, addressing stress is crucial.

As countermeasures for stress, well-balanced nutrition and dietary habits have been proposed [15]. It is known that some stressed individuals experience increased or decreased caloric intake. Some people tend to consume high-calorie foods, such as fast foods and snacks, under stress. There are two tendencies of hyperphagia and anorexia reported in humans [16]. Thus, the association between dietary habits and the stress response may vary among individuals. The effectiveness of stress amelioration has been widely investigated, from dietary styles to nutrient associations, with Mediterranean-style diets and DASH diets reportedly reducing feelings of anxiety and depression [17]; conversely, Western diets increase feelings of anxiety and depression [18]. Higher consumption of *n*-3 polyunsaturated fatty acids (PUFAs) is reported to be associated with a lower onset of depression [19,20]. Although the inhibition of depressive episodes correlates with PUFA consumption, it does not necessarily improve in a dose-dependent manner [21].

Based on these reports, dietary nutrients are metabolized by the gut microbiota, but differences in the types of gut microbiota may result in differences in metabolites, leading to varying effects on dietary stress. This study investigated the possible involvement of differences in gut microbiota in the relationship between diet and stress.

## 2. Materials and Methods

### 2.1. Study Design

The data used in this study were drawn from the Sukoyaka Healthy Survey conducted in 2019 and 2020 for adult males and females (aged 20 to 80 years) living in Ebetsu City, Hokkaido, and its suburbs [22]. In the investigation, information on background factors (age, gender, etc.) was obtained and a questionnaire (dietary habits and stress) administered for the subjects who provided informed consent. Fecal samples were collected by the subjects themselves using stool collection tubes (“Faeces tube with spoon”, Sarstedt, Nordrhein-Westfalen, Germany) with the addition of 10 mL of RNAlater (ThermoFisher, Waltham, MA, USA). This study was approved by the Bioethics Committee of Hokkaido Information University (approval date: 22 April 2019; approval number: 2019-04).

### 2.2. Questionnaire Data Acquisition and Processing

We used the FFQ (Food Frequency Questionnaire) data included in this observational research survey as the study data. The FFQ measures the frequency of intake (less than once a month, 1–2 times a month, 1–2 times a week, 3–4 times a week, 5–6 times a week, once a day, 2–3 times a day, 4–6 times a day, 7 times or more a day) for over 100 food items, as well as the amount consumed per eating occasion (less than 0.5 the standard amount, the standard amount, more than 1.5 times the standard amount). For the analysis, we used the daily intake amount (g/day) for each of the 13 food categories (cereals, potatoes, beans, green and yellow vegetables, other vegetables, fruits, mushrooms, seaweeds, seafood, meat, eggs, dairy products, alcohol), which was obtained by converting these FFQ data. Additionally, we adjusted the daily intake amounts (g/day) to reflect the intake per 1000 kcal of energy consumption (g/1000 kcal).

To measure the mental and physical responses of the subjects, the B-item of the Brief Job Stress Questionnaire [23] was used. In this questionnaire, the response scores of positive psychological responses (“vigor”) and negative psychological responses (“irritability”, “fatigue”, “anxiety”, “depression”) and negative physical responses (“somatic complaints”) were investigated.

### 2.3. Gut Microbiota

The collected fecal samples were used for whole-genome shotgun sequencing. The DNA extraction and sequence library preparation were conducted as outlined in [24]. Sequencing was performed on NovaSeq 6000 instruments using S4 flowcells (Illumina, San Diego, CA, USA). Raw sequencing reads were processed using fastp v0.20.0 [25].

Reads derived from the human genome were removed using BMTgger v3.101 (downloaded from http://ftp.ncbi.nlm.nih.gov/pub/agarwala/bmtagger/ (accessed on 1 April 2025)). Single-sample metagenome assembly was performed using MEGAHIT v1.2.9 [26] with default settings. Taxonomic profiling was performed using the Kraken ver.2 [27] and Bracken database ver. 2.7.0 [28].

### 2.4. Data Analysis

First, we conducted a cluster analysis of the study subjects based on the composition ratio of gut bacteria at the genus level. Then, the gut bacteria that characterize each cluster were identified at the genus level, and comparisons between clusters were made. To classify the gut microbiota of the subjects, hierarchical agglomerative clustering analysis was conducted based on the abundance ratios of genus-level strains. It has been reported that, in cluster analysis involving variables whose total is constant, such as composition ratios, using correlation coefficients as a measure of similarity can introduce bias [29]. In this study, we used Jensen–Shannon divergence (JSD) as the measure of similarity and chose the most conservative complete linkage. The number of clusters was determined using the Silhouette coefficient, Calinski–Harabasz index, and Davies–Bouldin index. Note that previous research employed JSD and partitioning around medoid (PAM) [7]. There is strong justification for using JSD to compare distributions themselves, but the choice of linkage in cluster analysis is not obvious. In this study, various linkages were compared, and complete linkage was selected as the method that yields the most conservative results in terms of cluster cohesion.

Regression analyses were conducted to determine whether there are differences in the relationship between dietary habits and mental and physical stress responses for each type of gut microbiota obtained as a result of the cluster analysis. The regression equation is as follows:(1)$${Stress\;response}_{i}=\alpha +\sum_{k=1}^{13}\beta_{j\left(i\right)k}{Food\;intake}_{ik}+\gamma^{\prime }x_{i}+\varepsilon_{i},$$
where ${Stress\;response}_{i}$ is one of the stress response scores of vigor, irritability, fatigue, anxiety, depression, and somatic complaints for subject i; ${Food\;intake}_{ik}$ represents the daily intake of food category k by subject i adjusted to reflect intake per 1000 kcal of energy intake, which has been converted to z-scores for the convenience of interpreting the regression coefficients; the subscript $j\left(i\right)$ of the coefficient parameter $\beta_{j\left(i\right)k}$ denotes the cluster number to which subject i belongs; and $x_{i}$ is the covariate vector for subject i, including a female dummy, age, a dummy variable indicating whether the individual lives alone, and stress response scores other than the dependent variable.

In regression Equation (1), the relationship between the intake of food category k and the stress response is represented by the coefficient $\beta_{j\left(i\right)k}$, which varies depending on the type of gut microbiota j of subject i. This formulation is equivalent to including the interaction terms between the dummy variables for membership in each gut microbiota cluster and the intake of each food category as explanatory variables.

## 3. Results

### 3.1. Gut Microbiota of Subjects

In the cluster analysis based on the genus-level composition ratio of the gut microbiota, Jensen–Shannon distance and complete linkage were used to calculate cluster cohesion indices within the range of two to nine clusters (Appendix A). As a result, four large clusters were formed (Appendix A). The largest sample size was in Cluster 4, with *n* = 596 (72.6%). Cluster 1 consisted of = 74 (9.0%), Cluster 2 of *n* = 64 (7.8%), and Cluster 3 of *n* = 87 (10.6%).

Strains with high abundance in each cluster were analyzed (Figure 1, Table 1). Cluster 1 had higher mean abundance ratios of Bacteroides (23.9%) and Bacteroides_B (17.5%). Cluster 2 had a higher abundance of Bifidobacterium (29.9%), as well as higher Bacteroides (12.5%) and Bacteroides_B (7.2%). Cluster 3 had a higher abundance of Prevotella (31.6%). Cluster 4 had higher levels of Bacteroides (15.3%) and Bacteroides_B (9.6%) and a higher abundance of Bacteroides_A (4.4%) than the other clusters.

The characteristics of the clusters were defined by Bacteroides, Bifidobacterium, Prevotella, and strains associated with the enterotype. Differences in strains between clusters were observed in other strains. In Cluster 1, Blautia (1.2%), Clostridium_M (3.1%), Faecalicatena (5.5%), Parabacteroides (4.0%), and Tyzzerella (1.4%) were significantly more prevalent than in other clusters, while Collinsella (0.7%) and Faecalibacterium (0.9%) were significantly less prevalent. Fuscicatenibacter (2.7%), Gemmiger (2.3%), Lactobacillus (1.2%), and Ruminococcus_E (1.2%) were significantly more prevalent in Cluster 2. Agathobacter (3.0%) and Megamonas (1.1%) were significantly more prevalent in Cluster 3. In Cluster 4, Agathobacter (3.1%), Alistipes (2.9%), Blautia_A (7.3%), Fuscicatenibacter (3.0%), Lachospira (1.6%), Roseburia (1.1%), and Ruminococcus_E (1.3%) were significantly more common.

Strain diversity was calculated using the Shannon index and Simpson index (Figure 2). Cluster 4 was the most diverse in both Shannon and Simpson indices. The Shannon index showed significantly higher diversity in the order of Cluster 4, Cluster 3, Cluster 2, and Cluster 1. In the Simpson index, there were no significant differences in diversity among Cluster 1, Cluster 2, and Cluster 3, but Cluster 4 had significantly higher diversity than the other clusters.

### 3.2. Regression Analyses

The results of the analysis showing how the daily intake of each of the 13 food categories relates to stress responses, categorized by the 4 types of gut microbiota composition, are presented in Figure 3. Figure 3 presents the estimation results of six regression equations corresponding to the six stress response items, displaying only the combinations of stress items and food classifications with coefficients that are significant at the 1% level. In each panel of each figure, the estimated coefficients $\beta_{j\left(i\right)k}$ from regression Equation (1) and their 95% confidence intervals are displayed, categorized by the types of gut microbiota composition. Note that the daily intake for each food category has been converted to z-scores. Accordingly, the coefficient β represents the change in stress response scores for a one-standard-deviation increase in intake. Although only the 1% significant coefficients for food category intake are shown here, the complete estimation results of the linear regression model, including those for other covariates, are available in Appendix A.

The relationship between food intake and stress is statistically significant at the 1% level in Clusters 1, 2, and 4. In Cluster 1 of the gut microbiota types, a higher intake of beans is associated with lower physical response score (β = −2.49, *p* = 0.001), greater meat intake is linked to a higher vigor-score (β = +1.16, *p* = 0.001), lower fatigue-score (β = −0.74, *p* = 0.010), and higher depression-score (β = +1.02, *p* = 0.005); and greater intake of eggs is linked to a higher depression-score (β = +1.08, *p* = 0.005).

In Cluster 2, meat intake is positively associated with fatigue-score (β = +0.44, *p* = 0.005), mushroom intake is positively associated with irritability-score (β = +0.77, *p* = 0.010), and the intake of vegetables other than green and yellow vegetables shows a positive association with fatigue-score (β = +1.60, *p* = 0.001). Additionally, in Cluster 4, green and yellow vegetables show a negative association with depression (β = −0.31, *p* = 0.005), while alcohol intake is positively associated with vigor-score (β = +0.28, *p* = 0.005).

## 4. Discussion

### 4.1. Gut Microbiota Clustering

To classify the gut microbiota of the subjects, hierarchical clustering was performed using the abundance ratios of genus strains, resulting in the formation of four clusters (Figure 1, Appendix A). Each cluster had distinct features of the gut microbiota. In the present categorization, Clusters 1, 2, and 4 showed a high abundance of Bacteroides and Bacteroides_B, with 89.4% of all subjects meeting these criteria. The other cluster included more Prevotella (10.6%). This finding was consistent with a previous report [9,30] concluding that the gut microbiota of Japanese adults predominantly consists of Bacteroides and Bifidobacterium. It has also been reported that some Japanese individuals have a certain number of Prevotella [4,31]. These results indicate that the gut microbiota characteristics presented in this report are not community-specific.

There was a higher ratio of Cluster 3 in males and a higher ratio of Cluster 2 in females. Cluster 3 had a higher abundance ratio of Prevotella, and Cluster 2 had a higher abundance ratio of Bifidobacterium. In prior research [4,31], it was reported that there was a difference in the composition ratio of bacteria between males and females in Japanese individuals, with a higher ratio of Prevotella in males and a higher ratio of Bifidobacterium in females. The results of this survey support these previously reported differences. However, differences in enterotype are known to be related to differences in dietary habits over the years, with the Bacteroides type tending to have higher protein and lipid intake and the Prevotella type having higher dietary fiber intake [9].

The data used in this study are limited to a specific region in Japan. However, Clusters 1 to 3, which were identified as a result of the analysis, are consistent with the findings of existing research. Conversely, Cluster 4 is characterized by diversity and can be considered a result unique to the data in this study.

### 4.2. Gut Microbiota, Diet, and Stress

Although gut microbiota, diet, and stress responses are mutually interconnected [32], this study focuses on the conditional nature of the relationship between diet and stress responses, mediated by gut microbiota. The analysis revealed differences in the relationship between food group intake and stress responses among clusters based on the genus-level composition ratio of gut microbiota.

Cluster 1 is characterized by a relatively high composition ratio of Bacteroides. Bacteroides is reported to have a negative correlation with dietary habits that involve relatively high fat intake, while having a positive correlation with dietary habits that involve relatively high dietary fiber intake [33,34]. In Cluster 1, physical complaints are negatively associated with bean intake, vigor is positively correlated with meat intake, fatigue is negatively associated with meat intake, and depression is positively linked to the intake of either meat or eggs. In relatively low-fat dietary habits, the intake of relatively high-protein foods may play a role in stress management.

Cluster 2 is characterized by a relatively high composition ratio of Bifidobacterium. Bifidobacteria are often highlighted as a target for prebiotics [35,36] or as probiotics [37,38] themselves. In Cluster 2, fatigue is positively associated with the intake of meat or vegetables other than green and yellow ones, while irritability is positively associated with the consumption of mushrooms. The analytical results regarding vegetables and mushrooms are difficult to interpret intuitively. Although not reported, the robustness of the results for other vegetables and mushrooms was confirmed even when the intake amounts by food category were converted into quantiles or when other covariates were restructured. Therefore, it might be best understood as reverse causation. That is, in Cluster 2, individuals who are irritable tend to consume more mushrooms, and those who feel fatigued tend to consume more of other vegetables.

In Cluster 4, characterized by relatively greater gut microbiota diversity, depression is negatively associated with green and yellow vegetable intake, while vigor is positively associated with alcohol consumption. If the diversity of gut microbiota is considered a measure of gut health, it can be assumed that individuals with better gut health have greater potential for stress management through green and yellow vegetables and alcohol.

In Cluster 3, characterized by a relatively high composition ratio of Prevotella, there is no statistically significant relationship at the 1% level between food intake and stress responses. Among the stress response items, anxiety does not exhibit a significant relationship with food intake at the 1% level in any cluster.

### 4.3. Limitations

The analysis in this study has several limitations and unresolved issues. First, we examined the relationship between dietary habits and stress responses based on gut microbiota types. However, the confounding factors controlled in the regression analysis were limited to gender, age, and a dummy variable for single-person households. In general, the evidence levels for causal inference range from the highest level, randomized experiments, to the lowest level, literature reviews. The analytical method in this paper, which employed regression analysis using cross-sectional data, falls into an intermediate evidence level. While more abundant confounding factors would raise the evidence level, due to sample size constraints in this study, we were forced to select confounding factors minimally. Furthermore, dietary habits, stress, and gut microbiota are mutually influential, making it impossible to eliminate simultaneous determination bias in the regression analysis conducted in this study. Additionally, it is important to distinguish between the long-term and short-term dynamics of gut microbiota, and the data used for analysis should include not only cross-sectional variation, but also time-series fluctuations. In these respects, the analysis in this study only addresses a very limited aspect of the inter-relationships among dietary habits, gut microbiota, and stress responses.

Secondly, the dietary habit data used in the study were obtained through a retrospective, self-administered questionnaire (FFQ) that evaluates dietary habits over the past year. Therefore, it is not possible to eliminate the influence of memory ambiguity or biases stemming from the respondent’s idealized self-perception of their dietary habits ([39] and its references).

## 5. Conclusions

In this study, the relationship between food intake and stress according to intestinal microbiota types was analyzed using data from intestinal microbiota, an FFQ, and occupational stress questionnaires. In the type with a relatively high proportion of Bacteroides, the intake of beans was shown to suppress physical complaints; in the type with relatively high diversity in intestinal microbiota, the consumption of green and yellow vegetables was found to reduce depression, and the intake of alcohol might help promote vitality.

## Figures and Tables

**Figure 1 nutrients-17-01388-f001:**
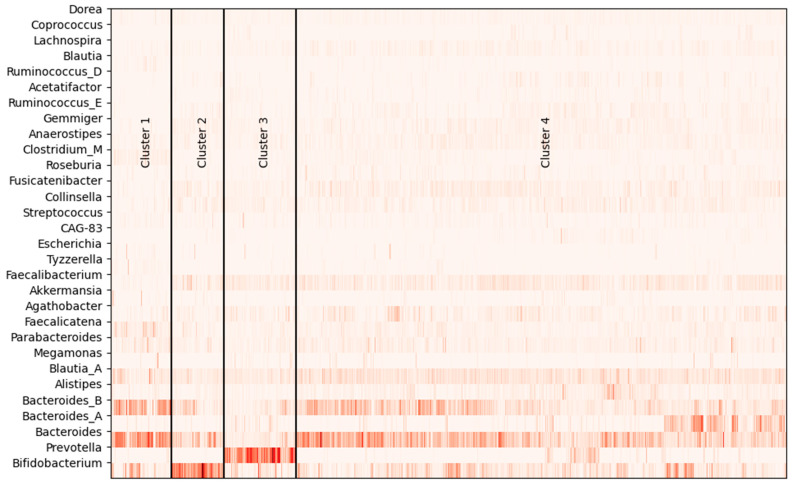
Clustering of gut microbiota of subjects. Illustration of the abundance ratios of the bacterial flora, plotted for the top 30 strains.

**Figure 2 nutrients-17-01388-f002:**
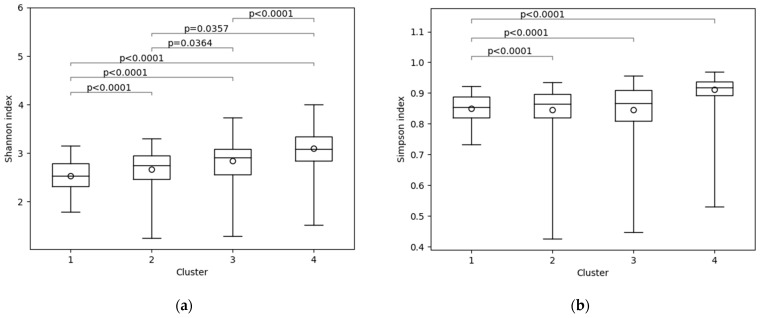
Multiple comparisons of diversity among clusters using the Dwass–Steel–Critchlow–Fligner all-pairs comparison test: (**a**) Shannon index; (**b**) Simpson index.

**Figure 3 nutrients-17-01388-f003:**
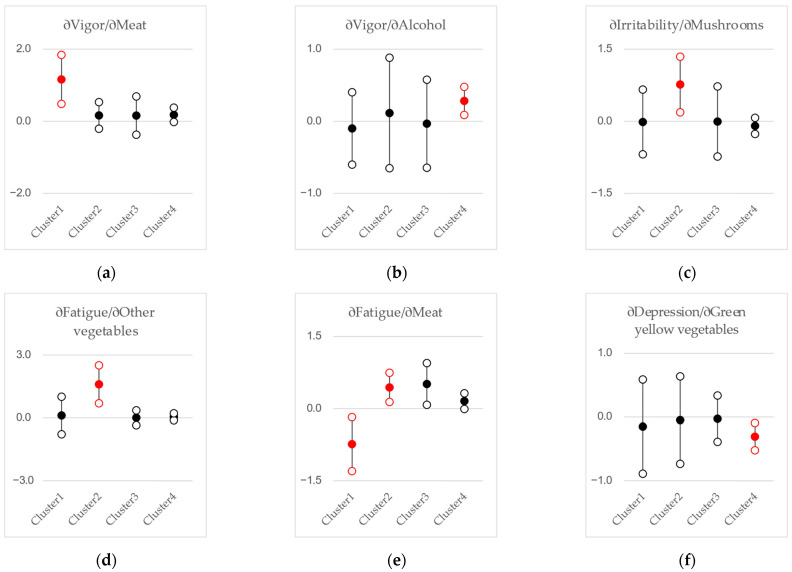
Estimated coefficients of regressor ${Food\;intake}_{ik}$ in Equation (1) for each regression of ${Stress\;response}_{i}$. For each of the six stress response scores, only the combinations of stress and food items with regression coefficients at a 1% significance level are shown, categorized by the 13 food item classifications based on gut microbiota clusters. (**a**) ∂Vigor/∂Meat; (**b**) ∂Vigor/∂Alcohol; (**c**) ∂Irritability/∂Mushrooms; (**d**) ∂Fatigue/∂Other vegetables; (**e**) ∂Fatigue/∂Meat; (**f**) ∂Depression/∂Green and yellow vegetables; (**g**) ∂Depression/∂Meat; (**h**) ∂Depression/∂Eggs; (**i**) ∂Physical/∂Beans. The filled circular markers represent the estimated coefficients, while the hollow markers indicate the upper and lower bounds of the 95% confidence interval. The red markers indicate that the regression coefficients differ significantly from zero at the 1% significance level.

**Table 1 nutrients-17-01388-t001:** Average values of major intestinal bacterial composition ratios by cluster.

Cluster	N	*Agathobacter*	*Alistipes*	*Anaerostipes*	*Bacteroides*	*Bacteroides_A*	*Bacteroides_B*
1	74	0.013	0.014	0.014	0.239	0.009	0.175
2	64	0.018	0.016	0.015	0.125	0.006	0.072
3	87	0.030	0.010	0.014	0.052	0.027	0.046
4	596	0.031	0.029	0.019	0.153	0.044	0.096
*p*-value *		<0.001	<0.001	<0.001	<0.001	<0.001	<0.001
Cluster	*Bifidobacterium*	*Blautia*	*Blautia_A*	*Clostridium_M*	*Collinsella*	*Dorea*	*Faecalibacterium*
1	0.076	0.012	0.064	0.031	0.007	0.009	0.009
2	0.299	0.002	0.052	0.005	0.026	0.010	0.046
3	0.055	0.002	0.050	0.006	0.022	0.010	0.045
4	0.074	0.003	0.073	0.009	0.022	0.010	0.048
*p*-value *	<0.001	<0.001	<0.001	<0.001	<0.001	0.001	<0.001
Cluster	*Faecalicatena*	*Fusicatenibacter*	*Gemmiger*	*Lachnospira*	*Lactobacillus*	*Megamonas*	*Parabacteroides*
1	0.055	0.006	0.005	0.007	0.000	0.006	0.040
2	0.020	0.027	0.023	0.008	0.012	0.000	0.023
3	0.017	0.017	0.012	0.010	0.000	0.011	0.021
4	0.020	0.030	0.019	0.016	0.000	0.004	0.029
*p*-value *	<0.001	<0.001	<0.001	<0.001	0.001	<0.001	0.001
Cluster	*Prevotella*	*Roseburia*	*Ruminococcus_E*	*Tyzzerella*			
1	0.006	0.005	0.002	0.014			
2	0.001	0.005	0.012	0.002			
3	0.316	0.008	0.006	0.002			
4	0.014	0.011	0.013	0.003			
*p*-value *	<0.001	<0.001	<0.001	<0.001			

* *p*-value: Kruskal–Wallis H-test.

## Data Availability

The data obtained from the “*Sukoyaka* Health Survey” are available in a publicly accessible repository managed by the DNA Data Bank of Japan (DDBJ) Japanese Genotype phenotype Archive at https://www.ddbj.nig.ac.jp/jga/index-e.html, accessed on 18 February 2025.

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
