# Peer review of "Relationship Between Dietary Habits and Stress Responses Exerted by Different Gut Microbiota"

_nutrients, 2025, doi:10.3390/nu17081388_

Round 1
Reviewer 1 Report
Comments and Suggestions for Authors
Reviewer Report
Title: Relationship between Dietary Habits and Stress Responses Exerted by Different Gut Microbiota
This manuscript explores the relationship between dietary habits, gut microbiota profiles, and stress responses among Japanese participants. The authors use cluster analysis to categorise gut microbiota and examine how these microbiota clusters moderate relationships between food intake and psychological stress indicators. The research question is relevant and timely, particularly given increasing interest in the gut-brain axis.
The strengths are that the study addresses an important topic linking dietary habits, microbiota composition, and mental health outcomes, specifically stress responses. The sample size is adequate, and the statistical analyses, including cluster and regression analyses, are thorough and appropriate. The exploration of distinct gut microbiota clusters provides valuable insights into the complexity of dietary interactions with mental health.
However, there are some concerns and recommendations for revision, despite the study’s strengths, several methodological and interpretative concerns need addressing.
A significant limitation is the reliance on self-report dietary measures (Food Frequency Questionnaires). Self-reporting can introduce recall bias and inaccuracies in estimating portion sizes, potentially affecting data accuracy and the validity of findings. The authors should explicitly acknowledge these limitations in their discussion and suggest incorporating objective dietary biomarkers in future research.
The cross-sectional design of the study limits causal interpretation. Although associations between dietary habits, microbiota composition, and stress responses are identified, causality and directionality remain unclear. Future research using longitudinal or interventional designs is recommended to strengthen these claims. The authors should explicitly discuss these limitations to avoid overstating their findings.
The choice of four clusters, although statistically supported, requires further justification in terms of clinical and biological significance. Supplementary data shows variability in silhouette and clustering indices. The authors should clearly articulate the rationale for selecting a four-cluster solution beyond statistical criteria and discuss the practical implications of these clusters.
Findings from this geographically and culturally specific sample (Hokkaido, Japan) limit generalisability to broader populations. The authors should at least mention potential differences in gut microbiota and dietary patterns across diverse populations and how these differences might impact the generalisability of their findings.
While overall well written, some manuscript sections, particularly methods and results, could be presented more concisely. Complex statistical information would benefit from clearer visual summaries, such as streamlined tables or figures, enhancing readability and reader comprehension.
Some more minor issues are that there needs to be consistent APA formatting or other accepted science style formatting throughout, particularly in references and figure/table captions. All terminology for food categories and microbiota strains throughout the manuscript should be standardised to enhance clarity and avoid confusion.
Overall, the manuscript makes a valuable contribution to understanding how gut microbiota may mediate dietary effects on stress responses. However, it requires revision before being considered for publication. Most of the points for revision can be addressed with more explanation and careful editing. Please accept the above points as guidance for the revision, after which it can be considered for publication.
Author Response
Comments 1:
This manuscript explores the relationship between dietary habits, gut microbiota profiles, and stress responses among Japanese participants. The authors use cluster analysis to categorise gut microbiota and examine how these microbiota clusters moderate relationships between food intake and psychological stress indicators. The research question is relevant and timely, particularly given increasing interest in the gut-brain axis.
The strengths are that the study addresses an important topic linking dietary habits, microbiota composition, and mental health outcomes, specifically stress responses. The sample size is adequate, and the statistical analyses, including cluster and regression analyses, are thorough and appropriate. The exploration of distinct gut microbiota clusters provides valuable insights into the complexity of dietary interactions with mental health.
However, there are some concerns and recommendations for revision, despite the study’s strengths, several methodological and interpretative concerns need addressing.
A significant limitation is the reliance on self-report dietary measures (Food Frequency Questionnaires). Self-reporting can introduce recall bias and inaccuracies in estimating portion sizes, potentially affecting data accuracy and the validity of findings. The authors should explicitly acknowledge these limitations in their discussion and suggest incorporating objective dietary biomarkers in future research.
Response1:
Thank you very much for your comments. We completely agree with your points, so we have added them as the limitation.
Comments 2:
The cross-sectional design of the study limits causal interpretation. Although associations between dietary habits, microbiota composition, and stress responses are identified, causality and directionality remain unclear. Future research using longitudinal or interventional designs is recommended to strengthen these claims. The authors should explicitly discuss these limitations to avoid overstating their findings.
Response 2:
Your comments are absolutely valid. We have added that as the limitation.
Comments 3:
The choice of four clusters, although statistically supported, requires further justification in terms of clinical and biological significance. Supplementary data shows variability in silhouette and clustering indices. The authors should clearly articulate the rationale for selecting a four-cluster solution beyond statistical criteria and discuss the practical implications of these clusters.
Response 3:
In Subsection 4.1, we compared the results with existing research and explained that clusters 1 to 3 represent commonly recognized classifications. Additionally, we noted that cluster 4 is a result unique to this paper.
Comments 4:
Findings from this geographically and culturally specific sample (Hokkaido, Japan) limit generalisability to broader populations. The authors should at least mention potential differences in gut microbiota and dietary patterns across diverse populations and how these differences might impact the generalisability of their findings.
Response 4:
Since this point is related to Comment 3, we addressed it together and referred to both the generality and specificity of the results of this paper in Subsection 4.1.
Comments 5
While overall well written, some manuscript sections, particularly methods and results, could be presented more concisely. Complex statistical information would benefit from clearer visual summaries, such as streamlined tables or figures, enhancing readability and reader comprehension.
Response 5:
Figures 3 through 15 were too detailed, so they have been revised to show only the results significant at the 1% level, consolidated into Figure 3.
Comments 6:
Some more minor issues are that there needs to be consistent APA formatting or other accepted science style formatting throughout, particularly in references and figure/table captions. All terminology for food categories and microbiota strains throughout the manuscript should be standardised to enhance clarity and avoid confusion.
Overall, the manuscript makes a valuable contribution to understanding how gut microbiota may mediate dietary effects on stress responses. However, it requires revision before being considered for publication. Most of the points for revision can be addressed with more explanation and careful editing. Please accept the above points as guidance for the revision, after which it can be considered for publication.
Response 6:
Basically, we have been using the Nutrients journal template and following the guidelines, but we found one part that did not comply with the instructions in the guidelines, so we have corrected it.
Reviewer 2 Report
Comments and Suggestions for Authors
Unclear Causality – Although the study identifies associations between gut microbiota, dietary habits, and stress responses, it does not establish causality. For example, it remains unclear whether stress also influences gut microbiota or if other confounding factors contribute to the observed relationships.
Potential Bias in Dietary Habit Measurements – The study relies on self-reported dietary data from questionnaires, which may be subject to recall bias or social desirability bias, potentially reducing the accuracy of the collected data.
Lack of Longitudinal Data – The study appears to be a cross-sectional study, meaning it does not track changes in diet, gut microbiota, or stress over time. This limits the ability to determine long-term effects or causal mechanisms.
Classification Method for Gut Microbiota May Affect Results – Participants were grouped based on gut microbiota abundance, but different classification standards could affect result stability and interpretation. The study does not specify whether standardized classification methods were used.
Omission of Other Influencing Factors – The study does not mention whether other potential factors affecting stress and gut microbiota—such as lifestyle (exercise, sleep), genetic differences, or psychological conditions—were considered. This omission may impact the accuracy of the findings.
Conclusion:
This study provides new insights into the relationship between gut microbiota, diet, and stress responses, reinforcing the concept of personalized nutrition. However, issues such as unclear causality, potential measurement bias, and a lack of longitudinal tracking remain. Future research should conduct long-term follow-up studies and incorporate additional influencing factors to enhance the reliability and applicability of the findings.
Comments on the Quality of English LanguageUnclear Causality – Although the study identifies associations between gut microbiota, dietary habits, and stress responses, it does not establish causality. For example, it remains unclear whether stress also influences gut microbiota or if other confounding factors contribute to the observed relationships.
Potential Bias in Dietary Habit Measurements – The study relies on self-reported dietary data from questionnaires, which may be subject to recall bias or social desirability bias, potentially reducing the accuracy of the collected data.
Lack of Longitudinal Data – The study appears to be a cross-sectional study, meaning it does not track changes in diet, gut microbiota, or stress over time. This limits the ability to determine long-term effects or causal mechanisms.
Classification Method for Gut Microbiota May Affect Results – Participants were grouped based on gut microbiota abundance, but different classification standards could affect result stability and interpretation. The study does not specify whether standardized classification methods were used.
Omission of Other Influencing Factors – The study does not mention whether other potential factors affecting stress and gut microbiota—such as lifestyle (exercise, sleep), genetic differences, or psychological conditions—were considered. This omission may impact the accuracy of the findings.
Conclusion:
This study provides new insights into the relationship between gut microbiota, diet, and stress responses, reinforcing the concept of personalized nutrition. However, issues such as unclear causality, potential measurement bias, and a lack of longitudinal tracking remain. Future research should conduct long-term follow-up studies and incorporate additional influencing factors to enhance the reliability and applicability of the findings.
Author Response
Comments 1:
Unclear Causality – Although the study identifies associations between gut microbiota, dietary habits, and stress responses, it does not establish causality. For example, it remains unclear whether stress also influences gut microbiota or if other confounding factors contribute to the observed relationships.
Response1:
Thank you very much for your comments. We completely agree with your points, so we have argued about them as a limitation.
Comments 2:
Potential Bias in Dietary Habit Measurements – The study relies on self-reported dietary data from questionnaires, which may be subject to recall bias or social desirability bias, potentially reducing the accuracy of the collected data.
Response 2:
Your comments are absolutely valid. We have added that as the limitation.
Comments 3:
Lack of Longitudinal Data – The study appears to be a cross-sectional study, meaning it does not track changes in diet, gut microbiota, or stress over time. This limits the ability to determine long-term effects or causal mechanisms.
Response 3:
Since this point is also related to Comment 1, we have referred to it together with that comment in the limitations.
Comments 4:
Classification Method for Gut Microbiota May Affect Results – Participants were grouped based on gut microbiota abundance, but different classification standards could affect result stability and interpretation. The study does not specify whether standardized classification methods were used.
Response 4:
The method commonly observed in existing research is the partitioning around medoids (PAM) method, which utilizes Jensen-Shannon distance (JSD). This paper adheres to the conventional approach in terms of using JSD as the distance measure but deviates by employing Complete linkage rather than PAM. In this study, Complete linkage was selected as the method that provides the most conservative results regarding cluster cohesion when compared to other linkage methods. While the rationale for using JSD as the distance measure is well-established, there is no particularly strong justification for the choice of linkage. This point has been added to the text.
Comments 5:
Omission of Other Influencing Factors – The study does not mention whether other potential factors affecting stress and gut microbiota—such as lifestyle (exercise, sleep), genetic differences, or psychological conditions—were considered. This omission may impact the accuracy of the findings.
Response 5:
We received feedback regarding the insufficient control of confounding factors in the regression analysis, which is closely related to Comment 1. Consequently, this point has been addressed alongside the response to Comment 1 in the limitations.
Comments 6:
Conclusion: This study provides new insights into the relationship between gut microbiota, diet, and stress responses, reinforcing the concept of personalized nutrition. However, issues such as unclear causality, potential measurement bias, and a lack of longitudinal tracking remain. Future research should conduct long-term follow-up studies and incorporate additional influencing factors to enhance the reliability and applicability of the findings.
Response 6:
We understand that you have summarized Comments 1 through 5 here. Please refer to the responses for each comment.